# The Importance of Oxidative Stress in Determining the Functionality of Mammalian Spermatozoa: A Two-Edged Sword

**DOI:** 10.3390/antiox9020111

**Published:** 2020-01-27

**Authors:** Robert J. Aitken, Joel R. Drevet

**Affiliations:** 1Priority Research Centre for Reproductive Sciences, Faculty of Science and Faculty of Health and Medicine, The University of Newcastle, Callaghan, NSW 2308, Australia; 2Hunter Medical Research Institute, New Lambton Heights, NSW, 2305, Australia; 3GReD Institute, INSERM U1103—CNRS UMR6293—Université Clermont Auvergne, Faculty of Medicine, CRBC building, 28 place Henri Dunant, 63001 Clermont-Ferrand, France; joel.drevet@uca.fr

**Keywords:** male infertility, oxidative stress, lipid peroxidation, sperm biology

## Abstract

This article addresses the importance of oxidative processes in both the generation of functional gametes and the aetiology of defective sperm function. Functionally, sperm capacitation is recognized as a redox-regulated process, wherein a low level of reactive oxygen species (ROS) generation is intimately involved in driving such events as the stimulation of tyrosine phosphorylation, the facilitation of cholesterol efflux and the promotion of cAMP generation. However, the continuous generation of ROS ultimately creates problems for spermatozoa because their unique physical architecture and unusual biochemical composition means that they are vulnerable to oxidative stress. As a consequence, they are heavily dependent on the antioxidant protection afforded by the fluids in the male and female reproductive tracts and, during the precarious process of insemination, seminal plasma. If this antioxidant protection should be compromised for any reason, then the spermatozoa experience pathological oxidative damage. In addition, situations may prevail that cause the spermatozoa to become exposed to high levels of ROS emanating either from other cells in the immediate vicinity (particularly neutrophils) or from the spermatozoa themselves. The environmental and lifestyle factors that promote ROS generation by the spermatozoa are reviewed in this article, as are the techniques that might be used in a diagnostic context to identify patients whose reproductive capacity is under oxidative threat. Understanding the strengths and weaknesses of ROS-monitoring methodologies is critical if we are to effectively identify those patients for whom treatment with antioxidants might be considered a rational management strategy.

## 1. Introduction

In a landmark paper published in 1943, McLeod [1] demonstrated that the incubation of human spermatozoa under conditions of high oxygen tension precipitated a loss of motility that could be reversed by the presence of catalase. The clear implication of these findings, that human spermatozoa are vulnerable to hydrogen peroxide attack was confirmed many years later when human spermatozoa were exposed to the mixture of reactive oxygen species (ROS) generated by the xanthine oxidase system, a known mediator of sperm oxidative stress in vivo and in vitro [2,3]. Exposure in vitro was found to induce a loss of fertilizing potential and, ultimately, motility, via mechanisms that could be completely reversed by concomitant exposure to catalase, which specifically catalyzes the conversion of hydrogen peroxide to oxygen and water, but was exacerbated by superoxide dismutase, which catalyzes the dismutation of superoxide anion to hydrogen peroxide [3]. Excessively high levels of spontaneous ROS generation were subsequently shown to be associated with the defective sperm function encountered in cases of human infertility [4,5]. The mechanism by which such oxidative stress induced defective sperm function was further shown to be linked to the capacity of these reactive oxygen metabolites to stimulate lipid peroxidation in spermatozoa [6,7,8]. These cells are particularly vulnerable to this process because they are richly endowed with polyunsaturated fatty acids, the double bonds of which facilitate the hydrogen abstraction process that initiates the peroxidation cascade [8]. The causative links between oxidative stress, lipid peroxidation and sperm function have subsequently been confirmed many times by independent groups for human spermatozoa [9,10,11,12,13] and the spermatozoa of every other mammalian species examined including the bull [14], stallion [15,16], pig [17,18], dog [19], cat [20], etc. This fundamental concept, pioneered by such luminaries as Jones and Mann [21], Gagnon [22] and Storey [6] is fundamental to our understanding of the aetiology of defective sperm function. Of course, it is not the only factor responsible for compromising the fertilizing potential of mammalian spermatozoa and it is important not to overstate the case. Many other factors are potentially involved in suppressing a function such as sperm motility, including excess intracellular calcium [23], dephosphorylation of phosphatidyl-inositol-3-kinase [24,25], exposure to motility-inhibiting proteins in semen, including seminal amyloid [26] and seminogelin [27], and exposure to sperm immobilizing antibodies [28]. However, the relative significance of oxidative stress as a major factor in the aetiology of defective sperm function has been clearly demonstrated in the few antioxidant studies that have included measures of lipid peroxidation (malondialdehyde [MDA]) in their evaluation schedule as well as a placebo control group [29]. In one such study, patients exhibiting asthenozoospermia (motility ≤ 40%) were treated with antioxidants (100 mg vitamin E) or placebo, 3 times a day for either 6 months or until the patient’s partner was diagnosed as pregnant, whichever came first. This study demonstrated that treatment with vitamin E significantly improved sperm motility while significantly reducing MDA levels in the spermatozoa in a manner that could not be replicated by administering a placebo formulation. A second, independent, placebo-controlled study drew the same conclusions; treatment with a combination of vitamin E and selenium for 3 months successfully decreasing MDA concentrations and concomitantly improving semen quality [30]. A further study which did not include a placebo control group but did measure MDA levels in seminal plasma, also confirmed the importance of lipid peroxidation in the pathophysiology of defective sperm function. Treatment for 90 days with an antioxidant preparation decreased seminal MDA and statistically improved all elements of the conventional semen profile including motility, concentration and morphology [31]. Another similar study, also concluded that antioxidant treatment (N-acetylcysteine in this case) for 3 months could simultaneously decrease seminal MDA levels while significantly improving sperm concentration and motility in the ejaculate [32]. Thus overall, there is a wide-ranging consensus based on thousands of patients that male infertility is associated with significantly elevated levels of MDA in semen which is, in turn, negatively correlated with levels of seminal antioxidant protection and key aspects of semen quality including sperm count, motility and morphology [33,34].

The apparent effectiveness of antioxidant therapy in suppressing oxidative stress while improving semen quality demonstrates the causative nature of these associations. It should be emphasized in passing that antioxidant trials that do not involve measurement of oxidative stress (sadly the majority) are essentially worthless for the very reason that not all defective sperm function is oxidatively induced. Clearly this is an area for further research involving properly controlled clinical trials with antioxidant formulations that are based upon a knowledge of sperm biochemistry and the bioavailability of the administered antioxidants within the male reproductive tract. This has been achieved in animal models where the cause of the infertility is unequivocally oxidative in nature. Thus, using the GPx5^−/−^ knockout model, mice can be generated in which the reproductive pathology observed is entirely dependent on the creation of a localized stress within the epididymis. Treatment of these animals with a carefully formulated antioxidant preparation resulted in a return of oxidative DNA-damaged spermatozoa to control levels. In the same study, the loss of fertility observed as a consequence of scrotal heat stress could also be completely abrogated by pre-treating the animals with the same antioxidant formulation [35].

So, there is an overwhelming volume of evidence indicating a cascade of causal interactions between oxidative stress, peroxidative damage to sperm lipids and DNA and impaired sperm function. A number of questions arise in the wake of such findings including: what are the situations in which sperm oxidative stress occurs? Is the oxidative stress systemic or localized? Where does the oxidative stress come from in terms of cell type and biochemical pathway? What species of ROS are involved? How can we best measure them? In the remainder of this review, we shall attempt to address these key issues.

## 2. What Are the Situations in Which Sperm Oxidative Stress Occurs?

### 2.1. Capacitation and Hyperactivation

There are multiple lines of evidence demonstrating that spermatozoa are professional generators of ROS because of the fundamental role these molecules play in the induction of sperm capacitation. Biochemically, one of the major pathways through which ROS promote capacitation is via the redox regulation of tyrosine phosphorylation. The significance of ROS in this context appears to apply to all mammalian species examined including man [36,37,38], rat [39], mouse [40], buffalo [41], bull [42] and stallion [43]. The mechanisms underpinning this redox effect on protein tyrosine phosphorylation are multifaceted and involve the stimulation of cAMP generation, the inhibition of tyrosine phosphatase activity and, the modulation of additional signal transduction cascades, including SRC (Rous sarcoma oncogene)—and ERK (Extracellular Receptor Kinase)—mediated pathways (Figure 1) [38,43,44,45]. It has also been demonstrated that the formation of oxysterols during sperm capacitation facilitates one of the hallmarks of sperm capacitation, the removal of cholesterol from the sperm plasma membrane [46]. Such a change is thought to enhance the fluidity of the plasma membrane, promoting critical intermolecular interactions that promote the development of a capacitated state.

One specific aspect of sperm biology driven by redox activity during sperm capacitation is the onset of hyperactivated motility. This work was pioneered by Gagnon and de Lamirande [47] who demonstrated that hyperactivation is a redox-mediated event, possibly reflecting the ability of ROS to induce tyrosine phosphorylation in the fibrous sheath of the sperm tail [48].

### 2.2. Capacitation and Sperm-Egg Interaction

Another redox-regulated aspect of sperm function that has received little attention is sperm-zona interaction. When human spermatozoa are incubated with ferrous ion promoters to enhance lipid peroxidation, there is, at levels of peroxidation that are still compatible with full viability and motility, a dramatic increase in the ability of these cells to bind to the zona pellucida [8]. This phenomenon has been used to enhance levels of fertilization in a mouse in vitro fertilization system in which the induction of sublethal levels of lipid peroxidation was shown to significantly increase the number of spermatozoa binding to the zona pellucida [49]. How the induction of peroxidative damage to the sperm plasma membrane enhances sperm-zona binding is unknown. It does not appear to involve a generalized increase in sperm adhesiveness because no amount of lipid peroxidation will enhance zona binding if the latter has been precoated with an anti-zona antibody (unpublished observations). During fertilization, the oocyte is also thought to generate ROS and may play an active role in the process of zona hardening, a proposed component of the block-to-polyspermy. Zona hardening is thought to be induced by peroxidases released during the cortical granule reaction enhancing the creation of molecular cross-links within zona pellucida with the aid of a hydrogen peroxide burst associated with fertilization [50]. This mechanism has been elegantly demonstrated in the case of the sea urchin fertilization envelope where the dual oxidase, Udx1, has been shown to generate the hydrogen peroxide associated with the hardening of this membrane [51]. The existence of an analogous process in mammalian oocytes seems likely; however, while these cells are known to contain a variety of oxidases capable of generating ROS [52], their role during fertilization remains largely unexplored. Whatever mechanisms are involved, it is clear that many aspects of sperm biology including tyrosine phosphorylation, cholesterol exclusion, hyperactivation and sperm-egg interaction are redox regulated.

As a result, capacitating spermatozoa in either the female reproductive tract or in vitro can be thought of as under physiological oxidative stress. In order to protect the spermatozoa during this critical time in their life history, sophisticated antioxidant defense mechanisms have developed, involving such key players as glutathione-*S*-transferase omega 2 and peroxiredoxin 6 [53,54]. As a consequence of such strategies, spermatozoa can engage in a redox-regulated capacitation cascade without fear of succumbing to the oxidative stresses involved. However, if a spermatozoon should fail to find an egg and the capacitation period is prolonged, even these defensive mechanisms are ultimately overwhelmed and the cell, now in a state of “over-capacitation”, enters a senescence pathway culminating in the enhanced release of ROS from the mitochondria and cell death [24]. Thus, capacitation and senescence can be regarded as components of redox-regulated continuum [55].

### 2.3. Inadequate Antioxidant Protection from Seminal Plasma

As a biological response to oxidative stress, seminal plasma has evolved one of the most powerful antioxidant fluids known to man, replete with a range of antioxidant enzymes and small molecular mass free radical scavengers that, combined, generate a level of total antioxidant power that is estimated to be 10× higher than blood [56]. This antioxidant cocktail includes catalase, superoxide dismutase (SOD), glutathione peroxidase, glutathione-S-transferase and peroxiredoxins as well as water-soluble (uric acid, hypotaurine, tyrosine, polyphenols, vitamin C, ergothionine and glutathione) and fat-soluble (all-*trans*-retinoic acid, all-*trans*-retinol, α-tocopherol, carotenoids and coenzyme Q10) scavengers [56,57,58]. There are now many studies indicating that there is a consistent negative relationship between the levels of antioxidant protection provided by seminal plasma and the appearance of male infertility as well as the incidence of miscarriage [33,59]. A recent development in this field has been the suggestion that the measurement of oxidation-reduction potential (ORP) in human semen samples is predictive of oxidative stress [60]. It will be of interest to determine whether such ORP measurements are providing diagnostic information over and above the measurement of total antioxidant potential. It will also be fascinating to determine whether the decreased seminal antioxidant protection observed in cases of male infertility is a result of local [61] or systemic [62] pro-oxidant factors.

### 2.4. Leucocyte Infiltration

The antioxidant properties of seminal plasma are particularly important in protecting spermatozoa from the oxidative stress created by infiltrating leukocytes, particularly neutrophils. Up to the point of ejaculation, spermatozoa in the seminiferous and epididymal tubules would have had little, if any, direct contact with activated phagocytic leukocytes. However following ejaculation, they become exposed to phagocytes, originating from the urethra and secondary sexual organs, which will be significantly elevated in cases of genital tract infection. These seminal leukocytes are in an activated state, generating free radicals and influencing seminal redox balance as reflected in several oxido-sensitive indices [63]. However, as long as seminal plasma is present, the spermatozoa are protected by the antioxidants contained therein [64,65]. However, when seminal plasma is removed, the leukocytes have free reign to attack the spermatozoa and limit their capacity for movement and fertilization. The presence of contaminating leukocytes, even in low numbers, in the washed sperm suspensions used for IVF therapy has been shown to have a profound impact on the success of this form of therapy [66]. Possible solutions to this problem include the incorporation of selected antioxidants into the sperm culture media used for IVF (e.g., glutathione, *N*-acetylcysteine, hypotaurine, etc.) or the targeted removal of the leukocyte population using magnetic beads or ferrrofluids coated with antibodies against the common leukocyte antigen, CD45 [67,68].

### 2.5. Cryostorage

Another situation in which oxidative stress is known to play a key role in the disruption of normal sperm function is cryopreservation. This relationship was first pointed out in the 1940s by the pioneering work of Tosic and Walton [69] who showed that the addition of an egg-yolk based extender to bovine spermatozoa resulted in a loss of motility that was dependent on the presence of a sperm amino acid oxidase generating high levels of hydrogen peroxide in response to the aromatic amino acids in egg yolk (particularly phenylalanine). In this scenario, the loss of membrane integrity on the part of non-viable spermatozoa enabled the aromatic amino acids in egg yolk to gain access to the intracellular oxidase and thereby generate quantities of hydrogen peroxide that could suppress the motility of viable cells in the immediate vicinity. A similar situation has been observed in ram [70] and stallion [71] spermatozoa. While human spermatozoa also contain an aromatic amino acid oxidase, it is not bound and becomes rapidly lost from cells if their plasma membrane integrity has been compromised [72]. One of the obvious solutions to this amino acid oxidase problem in domestic animals has been to incorporate catalase into the sperm cryopreservation medium [73].

While the damage induced in mammalian spermatozoa by cryopreservation is clearly multifactorial (involving, to various degrees, cold shock, osmotic disruption and intracellular ice crystal formation) oxidative stress is clearly a significant factor in the mediation of cryostorage injury. For this reason, there is intense interest in the use of antioxidants to promote cryosurvival. Recent studies have, for example, indicated that the specific activity of SOD in seminal plasma is related to the freezability of stallion, jackass and dog spermatozoa [74,75]. Loss of SOD from human semen samples is also thought to be a key factor in determining their ability to survive cryostorage [76]. Consistent with these observations, supplementation of cryopreservation media with both SOD and catalase has been found to enhance the post-thaw motility of human spermatozoa [77]. Exactly the same has been found for porcine and rooster spermatozoa cryopreserved in the presence of supplementary SOD and catalase [78,79]. SOD mimetics have also been shown to improve the cryopreservation of alpaca and ram spermatozoa and to improve subsequent blastocyst formation rates in the goat [80,81,82]. The combination of SOD mimetic and catalase was, predictably, more effective than mimetic alone [83].

In addition to the use of antioxidant enzymes to curtail oxidative stress during cryopreservation a large number of small molecular mass free radical scavengers have also been deployed for this purpose; entry of the terms “cryopreservation” and “antioxidants” into PubMed yields over 3000 references and the area has recently been thoroughly reviewed [84]. The antioxidants assessed included ascorbic acid, α-tocopherol, reduced glutathione, zinc oxide nanoparticles, zinc sulphate, resveratrol, quercetin, melatonin, L-carnitine, coenzyme Q, hypotaurine/taurine and butylated hydroxytoluene. In general, such antioxidants have a beneficial effect on sperm survival and functionality following cryopreservation although there is still much to be done to create the optimal antioxidant blend for protecting the spermatozoa of individual species.

### 2.6. Lifestyle Exposures

Another reason for spermatozoa to become oxidatively stressed relates to environmental and lifestyle exposures that either directly promote ROS generation or suppress levels of intrinsic antioxidant protection. A classic example is smoking. If men smoke heavily, their entire physiology is oxidatively stressed, as reflected by lower levels of antioxidants, such as ascorbate, in seminal plasma as well as enhanced levels of ROS generation by the spermatozoa. A 48% increase in seminal leukocyte concentration in male smokers also contributes to the level of redox stress experienced by the spermatozoa [85]. A particular pathological feature of cigarette smoking is that it generates a significant increase in oxidative DNA damage in spermatozoa as reflected by elevated levels of 8-hydroxy-2′-deoxyguanosine (8-OHdG) [86]. This increase in oxidative DNA damage may be attributable to both the above-mentioned depletion of antioxidant protection as well as the suppressive impact of cadmium (a critical constituent of cigarette smoke) on OGG-1 (8-oxoguanine-DNA glycosylase-1; the first enzyme in the base excision repair pathway responsible for removing 8-OHdG adducts from the genome) during spermatogenesis [87]. The high levels of 8-OHdG present in the spermatozoa of male smokers has, in turn, been associated with the increased childhood cancer rates observed in the offspring of male smokers [88,89]. Interestingly, we have identified a region of chromosome 15 (15q13–15q14) as a particular hot spot for oxidative DNA damage in human spermatozoa [90] and this is the very region of the genome that has recently been identified as contributing to the aetiology of acute lymphoblastic leukemia [91], one of the childhood cancers that we know to be associated with paternal smoking [92].

Obesity is another lifestyle factor responsible for inducing a state of oxidative stress in spermatozoa. This condition is associated with a generalized pro-inflammatory state associated with systemic oxidative stress, antioxidant depletion and oxidative sperm DNA damage [93]. Fortunately, this situation can be reversed, at least in mice, by the concomitant administration of micronutrients (zinc, selenium, lycopene, vitamins E and C, folic acid, and green tea extract) to counter the oxidative stress [94].

Different frequencies of electromagnetic radiation have also been suggested to induce oxidative stress in the male germ line. The impact of radiofrequency electromagnetic radiation (RF-EMR) has recently been reviewed and supports the general consensus that, like obesity, this form of radiation can induce ROS generation, reduce antioxidant protection and increase sperm DNA damage [95]. Moreover, a 2-step mechanism has been proposed to explain this phenomenon; in the first step, RF-EMR induces displacement of electrons from the electron transport chain in the inner mitochondrial membrane. These leaked electrons are immediately taken up by the universal electron acceptor, oxygen, to generate superoxide anion. In the second step, the latter dismutates into the powerful oxidant, hydrogen peroxide, which then initiates a lipid peroxidation cascade culminating in the generation of cytotoxic electrophilic aldehydes such as 4-hydroxynonenal (4-HNE). This aldehyde then immediately seeks out the nucleophilic centers of proteins in the immediate vicinity, alkylating the latter at the vulnerable amino acids, cysteine, lysine and histidine. The alkylation of proteins in the electron transport chain, such as succinic acid dehydrogenase, further disrupts the flow of electrons within the inner mitochondrial membrane leading to yet more ROS generation and exacerbating the state of oxidative stress, ultimately compromising both the functional competence of the spermatozoa and the integrity of their DNA [96,97]. The clinical significance of these findings can be found in several epidemiological studies demonstrating significant links between mobile phone usage and semen quality [95].

Importantly, it has also been clearly demonstrated that these RF-EMR effects are not mediated by increased heat. This is critical because raised intratesticular temperature is another means by which EMR can influence sperm quality. Such thermo-sensitivity is reflected in the location of the testes in a scrotal sac resulting in an intratesticular temperature of 34 °C–35 °C, 2–3 °C below the core body temperature of 37 °C. If intratesticular temperatures are raised by, for example, experimental cryptorchidism, then there is a rapid cessation of germ cell differentiation triggered by a sudden wave of Fas-mediated apoptosis and autophagy in pachytene spermatocytes and spermatids via mechanisms that can be reversed by the administration of antioxidants [98,99]. Similarly, the loss of fertility induced by warming the scrotal mouse testes to 42 °C for 30 min can be completely reversed by the administration of antioxidants [35]. Another situation in which raised intratesticular temperature is evident is varicocele. This condition is also known to be associated with a loss of sperm function and DNA integrity as a consequence of oxidative stress [100,101]. Surgical ligation of the left internal spermatic vein to correct the varicosity has been shown to reduce oxidative stress parameters and to improve sperm quality and the concomitant administration of antioxidants has been shown to facilitate this process [102,103]. As we enter an era of elevated ambient temperatures associated with global climate change, we might anticipate an increased incidence of male infertility, as well as an elevated risk of a paternally-induced mutations in the offspring, as a consequence of oxidative DNA damage to the spermatozoa. It is already known that exposure to elevated summer temperatures suppresses human seminal quality and that oxidative stress is a major mediator of this change [104,105]. Such exposure will be of particular significance to livestock species where exposure to elevated ambient temperatures might not only suppress fertility but also disrupt the true breeding of selected genetic traits [106].

### 2.7. Environmental Pollutants

Another situation associated with the creation of oxidative stress in the male germ line involves exposure to a wide range of environmental chemicals that are capable of directly stressing spermatozoa and inducing ROS generation. One example that has been recently highlighted is the preservative, parabens, which is present in many commercial aqueous products including vaginal lubricants. This mixture of parabenzoic acid esters is capable of stimulating the generation of mitochondrial and cytosolic ROS, inhibiting sperm motility and viability in a dose-dependent manner. The ability of individual parabens to activate ROS generation and induce oxidative DNA damage was related to alkyl chain length. At the concentrations used clinically, methylparaben inhibited sperm motility and affected cell viability while augmenting ROS production and oxidative DNA damage [107]. Similarly, the commonly encountered environmental toxicants, phthalate esters and bisphenol A (BPA), are known to possess a capacity to induce oxidative stress in spermatozoa by virtue of their ability to activate ROS generation [108,109]. In the case of BPA, the induction of oxidative stress is associated with a premature acrosome reaction, loss of sperm motility, reduced viability, disturbed ionic balance, and alterations of the sperm proteome [109]. The causative involvement of ROS in the pathological changes induced by exposure to BPA has been demonstrated by virtue of the ability of antioxidants (reduced glutathione and α-tocopherol) to reduce its pathological impact [110]. The related toxicants bisphenol F and bisphenol S have also been shown to disrupt reproductive function via an oxidative mechanism [111,112]. The sperm mitochondria appear to play a key role in the genesis of ROS in this model, as part of a self-perpetuating redox cycle that culminates in DNA damage and the induction of apoptosis [113]. In the case of phthalate esters, we have employed an invertebrate model (*Galeolaria caespitosa*) to show that these compounds not only induce high levels of oxidative stress in the spermatozoa but also have an impact upon the developmental normality of the embryo via an epigenetic mechanism that has not previously been reported [114]. In these studies, addition of dibutyl phthalate (DBP) to *Galeolaria* spermatozoa resulted in a highly significant dose-dependent inhibitory effect on fertilization and embryogenesis. At low levels of DBP exposure, fertilization could occur but the resulting embryos exhibited a disrupted pattern of cleavage and chromosome segregation resulting in the genesis of abnormal embryos. Such abnormalities were associated with the induction of oxidative stress in the spermatozoa associated with the suppression of SOD activity and formation of electrophilic lipid aldehydes (4-HNE). The latter were subsequently found to bind to the acrosome and sperm centriole. Since the latter is responsible for orchestrating cell division in the embryo, we propose that 4-HNE adduction has a significant impact on the ability of the sperm centrioles to serve as microtubule organizing centers in the zygote, impairing both the normal segregation of chromosomes during mitosis and impeding the cytoskeletal changes that underpin the process of cell division [110]. Whether similar mechanisms underpin the observed association between oxidative stress in the male germ line and developmental abnormalities in human embryos that culminate in repeated early miscarriage [59,115] is currently an open question that has not yet been addressed. It is known that ROS generation and DNA fragmentation are significantly elevated in the spermatozoa of female partners experiencing recurrent early pregnancy loss [116] however the importance of 4HNE adduction of sperm centriolar proteins in the aetiology of this condition is unknown.

### 2.8. Iatrogenic Stress and Sperm Preparation

A final scenario for the creation of oxidative stress in spermatozoa involves iatrogenic damage associated with the techniques we are currently using to separate spermatozoa from seminal plasma for IVF purposes. As indicated above, seminal plasma has evolved to protect spermatozoa from oxidative stress generated during the ejaculatory process when the spermatozoa are suddenly shifted from a low- to a high- oxygen tension environment contaminated with activated neutrophils and macrophages that are actively generating ROS. The most effective sperm isolation strategies are therefore those where the spermatozoa are isolated directly from semen rather than from a washed pellet, since in the latter situation, leukocytes are able to attack the spermatozoa without any of the protection normally afforded by seminal antioxidants. Swim up from semen, discontinuous density gradient centrifugation and electrophoretic isolation all fulfil this condition and generally generate high quality spermatozoa for insemination [117,118]. Discontinuous density gradient centrifugation has been reported to increase DNA damage in certain cases possibly because of the presence of transition metals such as iron and copper in the commercial colloidal silicon preparations used to create such gradients [119]. While susceptibility to the presence of such metals appears to vary from sample to sample [120] such impacts can be readily addressed by the incorporation of a metal chelators such as EDTA into the gradients [119].

## 3. What Types of ROS are Involved?

Given the importance of oxidative stress in determining the functionality of mammalian spermatozoa, it is reasonable to ask which forms of ROS are involved and how the offending species might be sensitively assessed for diagnostic purposes. The first point to make is that ROS, as their name implies, are extremely reactive molecules that are generated in all complex cellular systems and react readily not just with vulnerable substrates including lipids, proteins and DNA, but also with each other. Classically, superoxide anion has a half-life at physiological pH of a few seconds and is rapidly removed from biological systems via the action of SOD to create hydrogen peroxide. This process is biologically important because it converts a relatively inert, non-membrane permeant free radical anion into a membrane permeant oxidant that will readily interact with appropriate substrates. Superoxide anion will also interact with another free radical species generated by spermatozoa, nitric oxide (NO), to generate a powerful oxidant, the peroxynitrite anion (ONOO−). It has been proposed that the combined action of these oxidants, peroxynitrite and hydrogen peroxide, drive the oxidative processes responsible for the regulation of sperm capacitation [55]. The fact that scavengers of both hydrogen peroxide (catalase) and peroxynitrite (uric acid) can suppress capacitation in different species adds weight to this argument [121,122,123,124,125]. The complexity of reactive oxygen metabolites involved in regulating sperm functionality increases still further in the presence of transition metals which can catalyze the breakdown of lipid peroxides. This process generates lipid peroxyl and alkoxyl radicals that actively participate in the hydrogen abstraction process that promotes the lipid peroxidation chain reaction. The latter inevitably leads to the generation of small-molecular mass lipid aldehydes, such as 4 HNE, that bind to the mitochondria and stimulate yet more ROS generation in a self-perpetuating cycle [96,126].

The fundamental point here is that we must be careful not to oversimplify the chemistry responsible for the physiological oxidative drive to capacitation or the creation of pathological oxidative stress. There are likely to be many different radical and non-radical species involved in these processes originating from a wide variety of different sources. Mitochondrial ROS is clearly an important contributor [127] and recent data supporting a role for lipoxygenase in this process are exciting [128] and supported by the finding that unesterified unsaturated fatty acids such as arachidonic acid are potent triggers for ROS generation by human spermatozoa [129]. Since spontaneous ROS generation by human spermatozoa is correlated with their free polyunsaturated fatty acid content [130] this pathway may well be particularly important in the aetiology of excessive ROS generation by defective human spermatozoa. Other potential pathways of ROS generation by human spermatozoa include reduced nicotinamide adenine dinucleotide phosphate (NADPH) oxidases, particularly Nox 5 [131,132] and other poorly characterized plasma membrane redox systems, identified using the redox active probe, WST-1 [133]. There can be no doubt that spermatozoa are vulnerable to oxidative stress and that this susceptibility is exacerbated by the ability of spermatozoa to generate ROS from multiple sites and to increase this activity under conditions of stress. Resolving the extent to which the oxidative damage observed in cases of male infertility is a reflection of active ROS generation by the spermatozoa themselves (as when there is local abundance of free unesterified polyunsaturated fatty acids, for example) and/or a passive consequence of oxidative stress generated systemically (in response to obesity or cigarette smoking) is a key question that will have to be addressed in future studies.

## 4. How Can We Best Measure Oxidative Stress in the Germ Line?

If oxidative stress is such an important contributor to male infertility, it is critical that we develop robust methods to diagnose this condition within the infertile population. Where the oxidative stress is systemic, a direct measurement of lipid peroxides in seminal plasma seems to be the current method of choice. The measurement of MDA in seminal plasma has been found to reflect a variety of parameters associated with oxidative stress including DNA damage in spermatozoa, their capacity to generate ROS and both protein carbonyl and nitrotyrosine expression in semen [134]. As indicated above, seminal MDA has also been used as a diagnostic criterion in preparation for antioxidant therapy [29]. To date, there are no reports of 4-HNE levels in seminal plasma being used to diagnose oxidative stress even though this aldehyde is likely to be a more sensitive marker of lipid peroxidation than MDA [135].

Measurement of ROS generation by human spermatozoa is made particularly difficult because of the various oxygen metabolites involved and the low levels of ROS generation compared with contaminating cell types, particularly neutrophils (Figure 2). Extensive use has been made of luminol- and lucigenin-dependent chemiluminescence for diagnostic purposes and we have written extensively on the underlying chemistry of these reagents and their shortcomings [136,137]. For example, we have demonstrated that the one electron reduction of lucigenin required to generate chemiluminescence can be achieved by reductases, such cytochrome b5 reductase and cytochrome P450 reductase using NADH or NADPH as electron donors respectively. The diagnostic value of this probe therefore probably lies more in its ability to reflect the volume of residual cytoplasm retained by defective spermatozoa than their capacity for ROS generation via an NAD(P)H oxidase [138]. Exactly the same argument applies to the nitroblue tetrazolium assay which cannot be used as a probe for ROS under any circumstances [139]. Such shortcomings do not mean that potential ROS-generating entities such as NADPH oxidase have no role to play in the pathophysiology of spermatozoa, only that we currently lack the diagnostic methods needed to demonstrate what that significance might be.

Luminol is a different story and one that is often poorly understood. Luminol requires a one electron oxidation to create the luminol radical which is the *sine qua non* for chemiluminescence [140]. In granulocytes, the primary oxidation event depends on the action of myeloperoxidase and is probably mediated by the powerful oxidant, hypochlorous acid. In the case of spermatozoa, an intracellular peroxidase again appears to be responsible for mediating luminol-dependent chemiluminescence (LDCL) although the limited availability of peroxidase activity means that the spontaneous signal is low [141]. In order to improve the sensitivity of the assay, horse radish peroxidase has been used to sensitize the assay for the generation of extracellular hydrogen peroxide [141]. As the major source of extracellular hydrogen peroxide in the human ejaculate is contaminating leukocytes, the LDCL picture is characteristically dominated by these cells (Figure 2). If the leukocyte population is selectively removed using magnetic beads coated with antibodies against CD45 (the common leukocyte antigen) the luminol-peroxidase signal is reduced to background levels and no difference can be detected two high- and low-density Percoll sperm populations of differing quality [136]. However if, under these same leukocyte-free conditions, the spermatozoa are exposed to a reagent that will induce extracellular hydrogen peroxide release, such as the redox-cycling napthoquinone, menadione (Figure 2), then a very powerful luminol signal is generated [136]. Such results demonstrate that it is not so much the capacity of the luminol-peroxidase system to detect extracellular ROS that is open to question but rather the ability of this probe to detect the low levels of ROS released extracellularly by spermatozoa in the face of leukocyte contamination.

In order to generate the kind of sensitivity needed to detect differences in relative levels of spontaneous ROS generation associated with variations in sperm function, flow cytometry protocols have to be used. This methodology allows the operator to focus exclusively on the sperm population while any contaminating cells, such as precursor germ cells or leukocytes, can be carefully gated out. Under these conditions, leukocyte-free populations of spermatozoa from the high (functional sperm) and low (dysfunctional sperm) regions of discontinuous Percoll gradients can be readily distinguished on the basis of their reactivity with 3 probes (Figure 2), MitoSox red (mitochondrial ROS generation) dihydroethidium (total intracellular ROS generation) and H2DCFDA (dichlorodihydrofluorescein diacetate targeting intracellular oxidants such as hydrogen peroxide and, to a lesser extent, peroxynitrite). Although questions are occasionally asked about the specificity of these reagents for specific forms of ROS, the dynamic interchangeability of individual oxidants and free radicals means that such considerations are irrelevant in a diagnostic context. The fact is that these probes can detect differences in redox activity that are highly correlated with defective sperm function and therefore they have significant clinical value [136]. By contrast, use of more sophisticated techniques such as mass spectrometry that may be definitive but lack the sensitivity to detect the low levels intracellular ROS associated with spermatozoa are not helpful, no matter how impressive their powers of resolution.

## 5. Conclusions

In conclusion, the importance of oxidative stress as a major modulator of mammalian sperm function is incontrovertible. ROS have a positive role to play in driving the cascade of biochemical changes associated with sperm capacitation through their ability to control tyrosine phosphatase activity, stimulate cAMP generation and mediate the oxidation and ultimate release of cholesterol from the sperm plasma membrane. However, for obvious structural reasons these cells have very little defense against oxidative stress and are, thus, highly dependent on the powerful antioxidant properties of the fluids that surround them following their release from the germinal epithelium (seminiferous and epididymal tubule fluid and, for a short while at the moment of insemination, seminal plasma). If this extracellular antioxidant protection should be compromised for genetic, environmental or lifestyle reasons then the male germ line comes under oxidative attack. Oxidative stress may also be exacerbated by a variety of conditions that promote ROS generation by cells in the immediate vicinity (e.g., leukocyte infiltration secondary to infection) or by the spermatozoa themselves (exposure to RF-EMR, the stresses associated with cryopreservation, heat and exposure to a wide range of xenobiotics including bisphenol A, phthalate esters, parabens, etc). The complex combination of factors responsible for creating oxidative stress in the male germ line may well vary from patient to patient and is still not fully explored. Understanding the nature of these factors is significant because it will help guide the preconception care of such patients. Ensuring the functionality and genetic integrity of spermatozoa prior to conception is important, not just because it may promote the chances of conception but because it will minimize the genetic and epigenetic mutational load carried by the offspring and in so doing promote their long-term health trajectory.

## Figures and Tables

**Figure 1 antioxidants-09-00111-f001:**
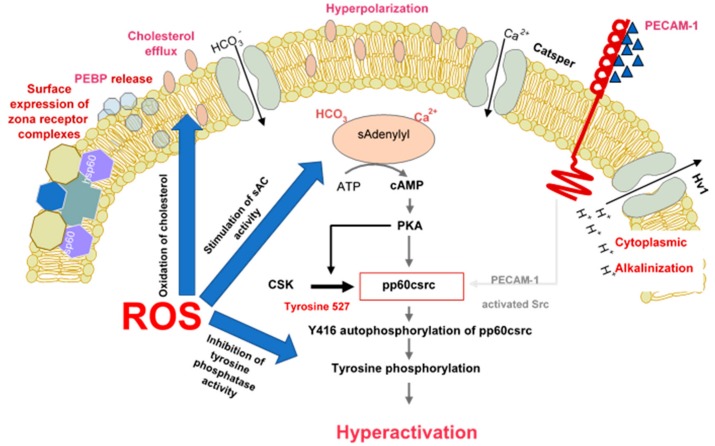
The role of reactive oxygen species in the induction of sperm capacitation. The latter is a complex process involving hyperpolarization of the sperm plasma membrane, cytoplasmic alkalinisation as a consequence of proton extrusion via the Hv1 proton channel, calcium entry via Catsper and a global increase in tyrosine phosphorylation mediated by cAMP. ROS (reactive oxygen species) are involved in several aspects of capacitation including cholesterol oxidation and extrusion, stimulation of soluble adenylyl cyclase (sAC) and suppression of tyrosine phosphatase activity.

**Figure 2 antioxidants-09-00111-f002:**
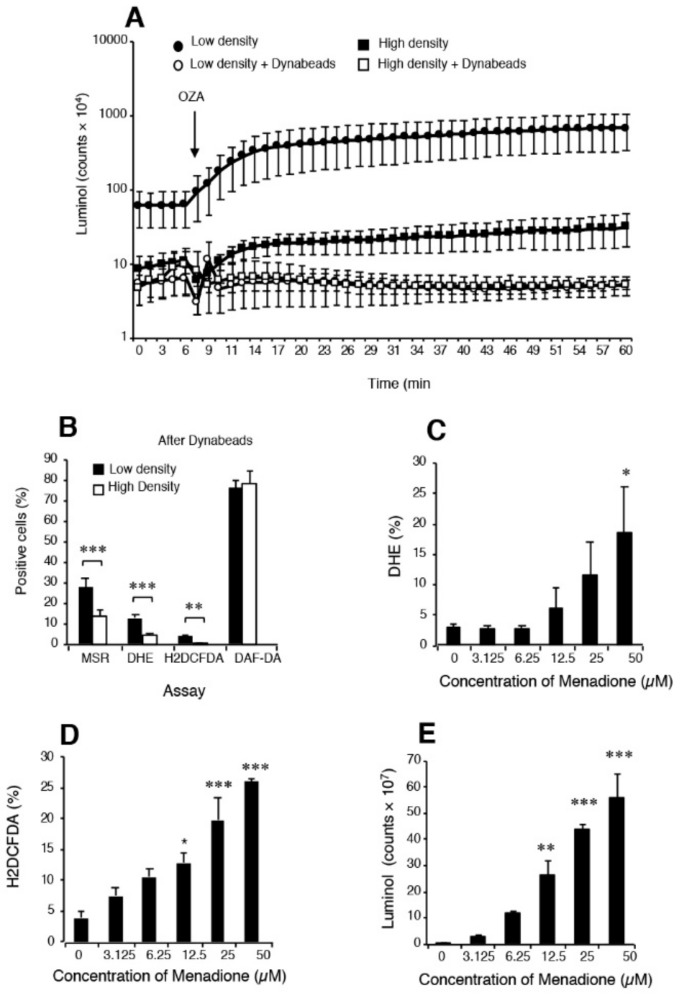
Analysis of spontaneous ROS generation by spermatozoa isolated from the high and low density regions of Percoll gradients. (**A**) Although luminol–horse radish peroxidase (HRP) dependent chemiluminescence could differentiate between high- (functional) and low- (dysfunctional) density sperm populations, such discrimination was completely lost when leukocytes were removed using CD45-coated Dynabeads; *n* = 6. (**B**) Following the removal of contaminating leukocytes, several ROS sensitive probes (MSR, DHE and H2DCFDA) but not the NO probe, DAF-DA, could discriminate the differences in sperm quality associated with high and low density Percoll populations. The most effective probe in this context was MSR, in keeping with the key role that mitochondria play in the aetiology of defective sperm function; *n* = 12. If leukocyte-free sperm suspensions were triggered to generate significant ROS using the redox cycling quinone, menadione (vitamin K), then several of the probes used in diagnostic andrology including MSR (not shown) (**C**) DHE, (**D**) H2DCFDA, and even (**E**) luminol/peroxidase could clearly detect a dose-dependent redox signal. Indeed, these dose-dependent analyses reveal that luminol and H2DCFDA were actually more sensitive than DHE in this regard. Overall, this analysis suggests that while probes such as luminol /peroxidase can clearly detect extracellular ROS generation, in practice their output is heavily influenced by the presence of contaminating leukocytes. Detecting differences in the spontaneous redox activity of the *spermatozoa* can, as indicated in panel B, only be achieved by flow cytometry using probes such as MSR, DHE and H2DCFDA (136). In panels (**A**,**E**), the chemiluminescence results are presented as counts per minute generated by the luminometer’s photomultiplier while in panels (**B**–**D**), the results are presented as the percentage of the sperm population exhibiting a positive response by flow cytometry. Abbreviations: MitoSOX Red (MSR), dihydroethidium (DHE), dichlorodihydrofluorescein diacetate (H2DCFDA), 4,5-diaminofluorescein diacetate (DAF-DA). OZA = opsonized zymogen to activate any phagocytic leucocytes in the cell suspension. Significance values: * *p* < 0.05; ***p* < 0.01 ****p* < 0.001.

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
