# Peer review of "The Importance of Oxidative Stress in Determining the Functionality of Mammalian Spermatozoa: A Two-Edged Sword"

_antioxidants, 2020, doi:10.3390/antiox9020111_

Round 1

Reviewer 1 Report

I read the review with title: The importance of oxidative stress in determining the functionality of mammalian spermatozoa: a two-edged sword, where the authors tried to very well demonstrate main aspects of action of oxidative stress to spermatozoa. They showen that reactive oxygen species generation is involved in whole process of sperm capacitaion and their proper function and also asummed up all of problems that can happen with right sperm function by the action of all king of oxidative stress.

I appreciate that autors did very extensive description of all situations in which can oxidative stress occur and that they have showen different methods how to measure the level of oxidative stress in spermatozoa. Itseems very important to know it and it is also very useful understanding the nature of complex action of all factors generating oxidative stress and influencing the mammalian sperm function.

I am glad I can fully recommend this review for publishing without any other comments or reminders.

Author Response

This referee appears to be fully supportive of publication without many changes being made. We are extremely grateful for this positive feedback. 

Reviewer 2 Report

In my view this is a valuable review, since a massive flow of papers on reactive oxygen species in different forms of pathology
have been recently registered. However, there are some flaws in the manuscript that must be either explained or amended - also in
our perception some critical reports in the field have been omitted.
In Introduction, line 39 after words in vitro - a key report on xanthine oxidase action
has been omitted and should be quoted considering particular links of pro-oxidants with defined pathological spermiograms
(Int J Androl., 1997; 20: 255-264). Furthermore chapter 2 should be subdivided into 8 subparts, namely I suggest
a) Sperm capacitation and hyperactivation should be separated from b) Sperm-egg interaction... since as now they are mixed
in one section but describing clearly different phenomena . Separate issue of sperm-egg interaction should be supplemented
with fundamental observation made in report published in Biol Reprod 1993; 49: 918-923. In present section 2.3 describing
leukocyte influence - there should be quoted issue of leukocyte presence in semen as well as proposed oxidosensitive index
for semen monitoring introduced in report published in J Reprod Immunol 2004; 62: 111-124.
Minor points - sentence between lines 361-366 is far too long. In Figure 2 the Y axis is not well described in most cases -
so what is measured under luminol (no units) DHA and H2DCFDA (percentage of what??). On X axis - why increasing
concentrations of Menadione have been applied??
In technical issues of measurements - ROS production by NADP/NADPH pump has been omitted - is it totally
redundant mechanism? It needs amendment.

Author Response

Point 1: In Introduction, line 39 after words in vitro - a key report on xanthine oxidase action has been omitted and should be quoted considering particular links of pro-oxidants with defined pathological spermiograms  (Int J Androl., 1997; 20: 255-264).

Response 1:  The reference has been inserted as requested - line 39.  

Point 2: Furthermore chapter 2 should be subdivided into subparts, namely I suggest  a) Sperm capacitation and hyperactivation should be separated from b) Sperm-egg interaction... since as now they are mixed in one section but describing clearly different phenomena. 

Response 2: These sections have been subdivided as suggested by the referee 

Point 3 Separate issue of sperm-egg interaction should be supplemented with fundamental observation made in report published in Biol Reprod 1993; 49: 918-923. 

Response 3: An additional section has been inserted on page 4 to incorporate this interesting observation - lines 157-167. 

Point 4: In present section 2.3 describing leukocyte influence - there should be quoted issue of leukocyte presence in semen as well as proposed oxidosensitive index for semen monitoring introduced in report published in J Reprod Immunol 2004; 62: 111-124. 

Response 4: This reference to the creation of oxidosensitive indices that reflect the response of the male genital tract to GTI, has been included in the revised text - lines 216 -219.

Point 5: Minor points - sentence between lines 361-366 is far too long.

Response 5: This sentence has been completely reconstructed as requested lines 488-508.  

Point 6:  In Figure 2 the Y axis is not well described in most cases - so what is measured under luminol (no units) DHA and H2DCFDA (percentage of what??). On X axis - why increasing concentrations of Menadione have been applied?? 

Response 6: Fig 2:  the Y axis of Fig 2 A  presents the results of this chemiluminescent assay in terms of counts per minute generated by the luminometer's photomultiplier. The flow cytometry results for probes such as DHE and H2DCFDA are presented as the percentage of the sperm population exhibiting a positive response.  This has been made clear in the revised Figure legend. 

Point 7: In technical issues of measurements - ROS production by NADP/NADPH pump has been omitted - is it totally redundant mechanism?

Response 7: I am not clear what 'the NADP/NADPH pump means'. If this refers to the use of lucigenin dependent chemiluminescence to measure ROS generation either spontaneously or in response to exogenous NADPH then, yes, I think the results obtained in such systems will be profoundly influenced by the presence of reductases. Lucigenin dependent activity may tell us more about cytoplasmic volume that ROS generation, as indicated in the  text.   However, this does not mean . that NADPH oxidases have no role to play in sperm pathophysiology - only that we lack definitive evidence to demonstrate their significance. This point has been made in the revised text - lines 586-589.